# REDUCING OVERSMOOTHING IN GRAPH NEURAL NETWORKS BY CHANGING THE ACTIVATION FUNCTION

## ABSTRACT

The performance of Graph Neural Networks (GNNs) deteriorates as the depth of the network increases. That performance drop is mainly attributed to over-smoothing, which leads to similar node representations through repeated graph convolutions. We show that in deep GNNs the activation function plays a crucial role in oversmoothing. We explain theoretically why this is the case and propose a simple modification to the slope of ReLU to reduce oversmoothing. The proposed approach enables deep architectures without the need to change the network architecture or to add residual connections. We verify the theoretical results experimentally and further show that deep networks, which do not suffer from over-smoothing, are beneficial in the presence of the "cold start" problem, i.e. when there is no feature information about unlabeled nodes.

## 1 INTRODUCTION

Graph Neural Networks (GNNs) utilize message passing or neighborhood aggregation schemes to extract representations for nodes and their neighborhoods. GNNs have achieved good results on a variety of graph analytics tasks, such as node classification (Zhang et al., 2018; Hamilton et al., 2017), link prediction (Liben-Nowell & Kleinberg, 2003; Zhang & Chen, 2018) and graph classification (Klicpera et al., 2020). As a result, they play a key role in graph representation learning. One of the most prominent GNN models is the Graph Convolutional Network (GCN) (Kipf & Welling, 2017), which creates node representations, by averaging the representations (embeddings) of its immediate neighbors. Several studies have shown that, the performance of GCNs deteriorates, when their architecture becomes deeper (Li et al., 2018).

The success of deep CNNs on many tasks, like image classification, naturally led to several attempts towards building deep GNNs for node classification (Kipf & Welling, 2017; Li et al., 2018; Xu et al., 2018). Increasing the model's depth (and the number of parameters it has), would allow more accurate representational learning to occur. Most of the existing approaches have failed to develop a sufficiently deep architecture that achieves good performance. Therefore, there is a need to design new models which can efficiently scale to a large number of layers. The aim of this paper is to investigate the contributing factors, that compromise the performance of deep GNNs and develop a method to address them.

The performance drop of deep GNNs, is associated with several factors, including vanishing gradients, overfitting, as well as the phenomenon called oversmoothing (Li et al., 2018; Xu et al., 2018; Klicpera et al., 2018; Wu et al., 2019). Oversmoothing has been shown to be associated with graph convolution, a type of Laplacian operator. Li et al. (2018) proved that applying that operator repeatedly, makes node representations converge to a stationary point. At that point, all of the initial information (i.e. node features' information) is lost through the Laplacian smoothing. Consequently, oversmoothing hurts the performance by making node features indistinguishable across different classes.

In this work, we address the oversmoothing problem in deep GNNs. We provide what is to the best of our knowledge the first study regarding the role of the activation function and the learning rate per layer of the model to oversmoothing and propose a new method to address the problem. We confirm our hypotheses both experimentally and theoretically. The new method is shown to prevent node embeddings from converging to the same point, thus leading to better node representations. We summarize our main contributions as follows.

•**Role of Activation Function in Oversmoothing**: We prove theoretically the connection between

the activation function and oversmoothing. In fact, we show the relation between the slope of ReLU and the singular values of weight matrices, which are known to be associated with oversmoothing (Oono & Suzuki, 2020; Cai & Wang, 2020). We have also verified our theoretical results experimentally.

•**Role of Learning Rate in Oversmoothing**: Our analysis on the effect of the slope of ReLU to oversmoothing has a direct extension to the learning rates used per layer of the network. We conducted further experiments to study the effect of tuning the learning rates, showing that this approach could also reduce oversmoothing, but it is less practical.

•**The power of Deep GNNs:** We have performed extensive experiments using up to 64-layer networks, tackling oversmoothing with the proposed method. We further show the benefits that such deep GNNs can provide in the presence of reduced information, such as in a "cold start" situation, where node features are available only for the labeled nodes in a node classification setting.

## 2 NOTATIONS AND PRELIMINARIES

### 2.1 NOTATIONS

In order to illustrate the problem of oversmoothing, we consider the task of semi-supervised node classification on a graph. The graph to be analysed is $\mathcal{G}(\mathbb{V}, \mathbb{E}, X)$, with $|\mathbb{V}| = N$ nodes $u_i \in \mathbb{V}$, edges $(u_i, u_j) \in \mathbb{E}$ and $X = [x_1, ..., x_N]^T \in \mathbb{R}^{N \times C}$ denotes the initial node features. The edges form an adjacency matrix $A \in \mathbb{R}^{N \times N}$ where edge $(u_i, u_j)$ is associated with element $A_{i,j}$. $A_{i,j}$ can take arbitrary real values indicating the weight (strength) of edge $(u_i, u_j)$. Node degrees are represented through a diagonal matrix $D \in \mathbb{R}^{N \times N}$, where each element $d_i$ represents the sum of edge weights connected to node i. During training, only the labels of a subset $V_l \in \mathbb{V}$ are available. The task is to learn a node classifier, that predicts the label of each node using the graph topology and the given feature vectors.

**GCN**, originally proposed by Kipf & Welling (2017), utilizes a feed forward propagation as:

$$H^{(l+1)} = \sigma(\hat{A} H^{(l)} W^{(l)}) \tag{1}$$

where $H^{(l)} = [h_1^{(l)}, ..., h_N^{(l)}]$ are node representations (or hidden vectors or embeddings) at the $l$-th layer, with $h_i^l$ standing for the hidden representation of node i; $\hat{A} = \hat{D}^{-1/2}(A+I)\hat{D}^{-1/2}$ denotes the augmented adjacency matrix after self-loop addition, where $\hat{D}$ corresponds to the degree matrix; $\sigma(\cdot)$ is a nonlinear element-wise function, i.e. the activation function, ReLU; and $W^{(l)}$ is the trainable weight matrix of the $l$-th layer. A generic ReLU function with slope $\alpha$ is defined as ReLU$(x) = max(\alpha \cdot x, 0)$.

### 2.2 UNDERSTANDING OVERSMOOTHING

GNNs achieve state-of-the-art performance in a variety of graph-based tasks. Despite their success, models like GCN (Kipf & Welling, 2017) and GAT (Velickovic et al., 2018) experience a performance drop, when stacking multiple layers. To a large extent, this is attributed to oversmoothing due to repeated graph convolutions. In their analysis Li et al. (2018), showed that graph convolution is a special form of Laplacian smoothing. In fact, they proved that the new representation of each node is formed by a weighted average of its own representation and that of its neighbors. This mechanism allows the node representations within each (graph) cluster, i.e. highly connected group of nodes, to become more similar and improves the performance on semi-supervised tasks on graphs. When stacking multiple layers, the smoothing operation is repeated multiple times leading to oversmoothing of node representations, i.e., the hidden representations of all nodes become similar, resulting in information loss.

### 2.3 DEEP GNN LIMITATIONS

Therefore, oversmoothing leads node representations to converge to a fixed point as the network's depth increases (Li et al., 2018). At that point, node representations contain information relevant to the graph topology and disregard the input features. Oono & Suzuki (2020) have generalized the idea in Li et al. (2018) by taking into consideration, that the ReLU activation function maps to a positive cone. They explain oversmoothing as the convergence to a subspace, instead of the

convergence to a fixed point. A similar approach is presented in (Cai & Wang, 2020), offering a different perspective to the oversmoothing problem using Dirichlet energy.

We now look closer to the proposal of Oono & Suzuki (2020), which we will use as a basis for our analysis. The main result in (Oono & Suzuki, 2020) is the definition of the convergence speed towards a subspace M, where the distance between node representations tends to zero. We denote as $d_M(X)$ the distance between the feature vector X and the subspace, where the oversmoothing is prevalent. When that distance approaches to zero it indicates that node representations have been over-smoothed. For this distance, Oono & Suzuki (2020) prove an interesting property.

**Theorem 1 (Oono & Suzuki (2020))** *Let the largest singular value of the weight matrix $W_{lh}$ be $s_{lh}$ and $s_l = \prod\limits_{h=1}^{H_l} s_{lh}$, where $W_{lh}$ is the weight matrix of layer $h$ and $H_l$ be the network's depth, following the notation of the paper. Then it holds that $d_M(f_l(X)) \leq s_l d_M(X)$ for any $X \in \mathbb{R}^{N \times C}$, where f($\cdot$) is the forward pass of a GNN layer (i.e. $\sigma(AXW_{lh})$).*

The above theorem indicates, that the deeper the network the smaller the distance of node representations from the subspace M. If the maximum singular values are small then node representations asymptotically approach M, for any initial values of node features. Extending the above theorem, the authors conclude to the following estimate about the speed of convergence to the oversmoothing subspace.

**Corollary 1 (Oono & Suzuki (2020))** *Let $s = sup_{l \in N_+} s_l$. then $d_M(X^{(l)}) = O((s\lambda)^l)$, where l is the layer number and if $s\lambda < 1$ the distance from oversmoothing subspace exponentially approaches zero. Where $\lambda$ is the smallest non-zero eigenvalue of I - $\hat{A}$.*

According to the authors, sufficiently deep GCN will certainly suffer from oversmoothing under some conditions (details can be found in Oono & Suzuki (2020)). We build upon this result, aiming to develop a consistent approach that reduces oversmoothing and enables deep architectures.

### 2.4 NORMALIZATION IN DEEP NEURAL NETWORKS

Of relevance to oversmoothing are also methods that perform normalization for deep neural networks (Ioffe & Szegedy, 2015; Ba et al., 2016). In such methods, the output of each neuron is normalized, in order to keep a portion of the initial feature variance. Other recent work proposed Self Normalized Networks (SNN), that use a different activation function (SeLU) (Klambauer et al., 2017). These models perform self normalization inside each neuron, in order to keep the variance between consecutive layers stable. They have managed to enable deep Fully Connected models and achieve good performance.

A recent work regarding normalization in GNNs is the Pairnorm method (Zhao & Akoglu, 2020), which aims to keep constant the total pairwise distance between node representations. Compared to SNNs, Pairnorm performs normalization needing a constant value as hyper-parameter determined per dataset, while SNNs manage to normalize the output utilizing their activation function, which can be used in a family of neural networks. The activation function proposed in (Klambauer et al., 2017) (i.e. SeLU) has a saturating region to reduce the variance of data and a slope slightly greater than 1 in order to increase the data variance, when needed.

## 3 UNDERSTANDING AND DEALING WITH OVERSMOOTHING

Using the mathematical definition of oversmoothing in Oono & Suzuki (2020), in this section we establish a connection with the training process of GNNs. In particular, we analyse the role of the activation function and the learning rate and we propose modified versions of GNNs to address the issue.

### 3.1 THEORETICAL ANALYSIS

We start by establishing the connection between oversmoothing and variance reduction of node representations. Consider oversmoothing as the convergence to a subspace, where node representations

are almost the same. In that particular subspace, the initial variance of feature vectors has been massively reduced. This unwanted oversmoothing effect is 'harsh', due to the fact that ReLU performs a zero mapping, when the input is negative. We use the term 'harsh' to indicate the smoothing case, where representations are mapped to a single point (i.e. zero), instead of converging to a subspace. We use the notion of 'converging to a subspace' following the oversmoothing analysis presented in Oono & Suzuki (2020), where the subspace M (where oversmoothing is prevalent) is defined.

We will now show how the slope of the ReLU activation function affects oversmoothing, starting with some important assumptions: (a) the non-exploding gradients assumption, and (b) ReLU's probability not to output zero. We use these assumptions to extract bounds on the weights of the network and then use these bounds to determine the relationship between the largest singular value and the elements of each weight matrix. Given that relationship, we connect our analysis with existing literature regarding oversmoothing through Theorem 1.

**Assumption 1** *For each layer l there exists a number $G_l$ which is the upper bound to the gradients of the output of the subsequent layer (l+1) with respect to the weight elements of $W^{(l)}$, i.e.*

$$\frac{dO^{(l+1)}}{dw_{old_{i,j}}^{(l)}} \leq G_l, \quad \forall w_{old_{i,j}}^{(l)} \tag{2}$$

*where $O^{(l+1)}$ is the output of (l+1)-th layer.*

This assumption needs to hold in all cases, in order to avoid the exploding gradient case, that would not allow the learning process to converge.

**Assumption 2** *The probability, that the output of the ReLU function not equals zero at layer $l$ is independent of the outputs of the previous layers, in a feed-forward ReLU neural network, i.e.*

$$P(ReLU(h^{(l)}) \neq \vec{0}|\forall h^{(j)}, j < l) \leq p, \quad p \in [0, 1) \tag{3}$$

*where $h^{(l)}$ is the node representation at layer $l$, ReLU is applied piece-wise on that representation and $\vec{0}$ is the all zero vector.*

Following the results of Lu et al. (2019) we get that $P(ReLU(h^{(l)}) \neq \vec{0}|\forall h^{(j)}, j < l) = 1/2$ for feed-forward networks (FFs), on the condition that the output of the previous layers is positive. That condition does not necessary hold in GNNs, due to the role of the adjacency matrix in the aggregation scheme. In FFs, when the output of a layer is zero, so are the outputs of all subsequent layers. In GNNs there might be cases, where the output of layer $l$ (a node representation) is zero but the aggregation using the adjacency matrix produces a non-zero representation in layer $(l+1)$. Apart from this difference, GNNs can be considered a special case of FFs, with the proper wiring due to the adjacency matrix aggregation. Therefore, the above assumption holds also for GNNs.

**Lemma 1** *For a network of depth dep the total gradient reaching to the l-th layer $\left(i.e., \frac{dJ}{dw_{old_{i,j}}^{(l)}}\right)$ in order to update $W^{(l)}$ is bounded by:*

$$\frac{dJ}{dw_{old_{i,j}}^{(l)}} \leq \alpha^{(dep-l)} \cdot G_l \tag{4}$$

*Where J is the model's loss function (i.e. Cross Entropy), $\alpha$ stands for the ReLU slope and $G_l$ is the upper bound of gradients of the output of the subsequent layer (l+1) with respect to the weight elements of $W^{(l)}$.*

The proof of Lemma 1 is shown in Appendix A and is based on the chain rule for backpropagation. Given the derivative of the ReLU function and the upper bound $G_l$ of Assumption 1, we can derive the bound of Equation 4. To compute the gradient with respect to weight element $w_{i,j}^{(l)}$ of layer $l$, we repeatedly differentiate nested ReLU functions, leading to a product of their slopes (or zero). In the final differentiation step, we get the gradient of the output of $(l+1)$-th layer with respect to $w_{i,j}^{(l)}$, which is bounded by $G_l$.

**Lemma 2** *While model's loss, through gradients, flows backwards some weight elements do not receive updates, because we have dying ReLUs (i.e. ReLUs outputting zero) (Lu et al., 2019). The total number of weight elements getting updated at layer l is bounded by:*

$$\#\{w_{i,j}^{(l)}\} \leq p^{(dep-l)} \cdot d^2 \tag{5}$$

*where d is the largest of the two dimensions of $W^{(l)}$, i.e. there are at most $d^2$ elements in $W^{(l)}$, if it is a square matrix.*

The proof of Lemma 2 appears in Appendix B. Given Lemma 1, the gradient flowing backwards, with respect to a weight element $w_{i,j}^{(l)}$ contains a product of ReLU derivatives. In order for $w_{i,j}^{(l)}$ to get an update, all ReLU derivatives need to be non-zero, otherwise the gradient will be zeroed and no update will be applied to $w_{i,j}^{(l)}$. Additionally, due to Assumption 2 weight elements located in the lower weight matrices tend to receive fewer updates. This is due to the fact, that the further the gradient flows backwards the more ReLU derivative factors appear in it. Based on Assumption 2, the probability of all of them to be non-zero decreases.

Regarding the singular values (denoted by $s_l(\cdot)$) of a weight matrix $W^{(l)}$ at the $l$-th layer, the largest of them is given by: $max(s_l(W^{(l)})) = ||W^{(l)}||_2 \leq ||W^{(l)}||_F = \sqrt{\sum |w_{i,j}^{(l)}|^2}$, where $|| \cdot ||_F$ is the Frobenius norm and $|| \cdot ||_2$ is the spectral norm. Using the general weight updating rule $\left(\text{i.e., } w_{new_{i,j}}^{(l)} = w_{old_{i,j}}^{(l)} + \eta \cdot \frac{dJ}{dw_{old_{i,j}}^{(l)}}\right)$, where J is the model's loss and $\eta$ is the learning rate, we arrive at the main theorem of this work.

**Theorem 2** *The upper bound of the largest singular value of the weight matrix $W^{(l)}$ at layer l for a GNN model, utilizing a ReLU activation function, depends on the slope of the function. That bound is given per layer and shows the effect of each iteration of updates on the weight matrix. We denote with $W_{old}$ and $W_{new}$ the weight matrices before and after the update during an iteration of the training process respectively.*

$$max(s_l(W_{new}^{(l)})) \leq ||W_{old}^{(l)}||_F + \sqrt{3} \cdot p^{\left(\frac{dep-l}{2}\right)} \cdot d \cdot \alpha^{(dep-l)} \cdot B_l \tag{6}$$

*where $B_l = \eta \cdot G_l$, dep is the network's depth, d is the largest dimension of the $W^{(l)}$ matrix, p is the upper bound of the probability of ReLU not to output zero and $\alpha$ is ReLU's slope.*

The proof of Theorem 2 appears in Appendix C and uses the upper bound of the largest singular value by the Frobenius norm, expanded according to the weight update rule. Separating weight elements into two sets (updated and not updated) we identify the gradient values responsible for the weight updates. These values are bounded (Lemma 1), as is the number of updated elements (Lemma 2), leading to Equation 6. The resulting upper bound shows the connection of the slope of ReLU and the upper bound of the probability of ReLU not to output zero with the largest singular value of the weight matrix. That largest singular value is connected with the oversmoothing problem and as we mentioned above could act as a resource to reduce it.

## 3.2 ALLEVIATING OVERSMOOTHING

We transfer the idea of SeLU to GNNs, focusing on the part that increases the variance, because the repetition of the Laplacian operator acts as a variance reducer. In order to avoid oversmoothing in deep GNNs using this approach, we need to identify a 'sweet' spot, where variance reduction from graph convolution counteracted as needed by the slope of the activation function. Typically in GNNs, ReLU is used with a slope value $\alpha = 1$. As a result, the second exponential factor in Equation 6 can be ignored (it is always equal to 1). This pushes the bound for the largest singular value of weight matrices towards a fixed low value, when the architecture of the network gets deeper ($dep$ increases), and especially at the lower layers (small $l$) of the network. This is due to the first exponential factor of (6) converging to zero for large values of $dep$.

The restriction of the largest singular values to low values, increases the speed of convergence to the oversmoothng subspace, as stated in Corollary 1 from Oono & Suzuki (2020). According to the corollary, the speed depends on $\lambda$ and s. The former parameter ($\lambda$) is a property of the data (largest eigenvalue of the adjacency matrix), while the latter (s) is the product of the largest singular values

of the weight matrices.

Realizing the importance of the slope of ReLU in the training process, we move on to propose a modification of the activation function that reduces oversmoothing. Following (Lu et al., 2019), we proceed our analysis with $p = 1/2$ (see Assumption 2), which simplifies calculations. In particular, we observe that a slope of 2 makes the second exponential factor prevail over the first one, leading to a new combined factor of $2^{\left(\frac{dep-l}{2}\right)}$. This in turn increases the upper bound for the largest singular value, restricting the influence of the layer index ($l$) and depth ($dep$) in Equation 6. It is worth noting, that the upper bound should not be constant across all layers (which could be achieved by setting a corresponding value for the slope), because different layers of the network have different roles and the goal of a GNN is to bring intra-class representations close, while keeping inter-class representations apart.

Using the results of (Oono & Suzuki, 2020) we can further connect the choice of slope for the activation function to the speed of convergence to the oversmoothing subspace. Hence, the proposed method increases the bound for each largest singular value, which in turn decreases the speed of convergence to the oversmoothing subspace, according to Corollary 1.

### 3.3 MODIFYING THE SLOPE OF ReLU: LIMITATIONS

Oono & Suzuki (2020) have proved that any GNN deep enough will eventually reach the oversmoothing subspace. In theory, this also holds for our method, when the GNN gets very deep. Could this be avoided by increasing the slope of ReLU further? Unfortunately not, as we cannot make the slope too large. It is known, that by increasing the slope of ReLU too much, one may face the problem of exploding gradients (Bengio et al., 1994), that impedes the learning procedure. In fact, if we increase the slope of ReLU too much, the upper bound in Equation 6 becomes too loose and the performance degrades. Our proof suggests, that a slope equal to 2 avoids oversmoothing and our experiments have shown that it also avoids the exploding gradients problem.

### 3.4 MODIFYING THE LEARNING RATE, INSTEAD OF THE SLOPE OF ReLU

Instead of changing the slope of ReLU, we could opt to modify the Learning Rate (LR) per layer in order to increase the upper bound in Equation 6. In particular, a different learning rate ($\eta$) per layer leads to a different $B_l$ (= $\eta \cdot G_l$) per layer, counteracting the problematic first exponential factor of Equation 6.

However, determining the right LR per layer is a non-trivial task. Intuitively, LR should be larger in the lower layers and smaller in the upper ones. Keeping the slope equal to 1, the first problematic exponential term is smaller (note $p < 1$) in the lower layers reducing the value of the upper bound. To avoid this, one needs to tune LR carefully. Additionally, LR affects heavily the learning process and large LR values might lead to oscillations and poor learning performance. Modifying the slope and the learning rate combined led to negligible improvements. Preliminary results with modified LR values are shown in Appendix D.

### 3.5 WHY DO WE NEED DEEP GNNS?

Oversmoothing is only a problem for deep GNNs, leading to the question of whether and when deep GNNs are really needed. Most of the existing benchmark datasets for various graph analytics tasks do not seem to justify the need for deep networks. Due to their homophilic nature, useful information for each node resides in its close neighbors (usually 2 or 3 hops away). A task where deeper architectures could be needed is the "cold start" problem; namely the situation where many node features are missing. The problem of missing features is called "cold start", because it resembles the situation of a new product/user arriving to a recommender system. The system has no prior knowledge about the new arrival, yet has to make some recommendations. In this scenario, the hope is that deeper GNNs could recover features from more distant nodes, in order to create informative representations.

## 4 EXPERIMENTS

### 4.1 EXPERIMENTAL SETUP

**Datasets:** Aligned to most of the literature, we use four well-known benchmark datasets: *Cora, CiteSeer, Pubmed* and the less homophilic dataset *Texas*. The statistics of the datasets are reported in the Appendix. For *Cora, CiteSeer* and *Pubmed* we use the same data splits as in Kipf & Welling (2017), where all nodes except the ones used for training and validation are used for testing. For *Texas* we use the same splits as in (Pei et al., 2020).
**Models:** We utilize two different GNN models as our base models for the proposed methodology; namely GCN (Kipf & Welling, 2017) and GAT (Velickovic et al., 2018). We compare the results of these models with and without the modification of the slope of ReLU for varying number of layers. We do not compare against methods utilizing residual connections, because the aim of this work is to show that oversmoothing could be avoided without "short-circuiting" initial information to latter layers. We use residual GNNs as a baseline in the Extended Experiments section, Appendix F.
**Hyperparameters:** We set the number of hidden units (of each layer) of GCN and GAT to 128 for both models across all datasets. The $L_2$ regularization is set with penalty $5 \cdot 10^{-4}$ for both models and learning rate set to $10^{-3}$. We vary the depth between 2 and 64 layers. The number of attention heads for GAT is set to 1.
**Configuration:** Each experiment is run 10 times and we report the average performance over these runs. We train all models for 200 epochs using Cross Entropy as a loss function. For our experiments we used an RTX 3080ti GPU.

### 4.2 EXPERIMENTAL RESULTS

**Reducing oversmoothing:**
Table 1 presents the classification performance of the two base models (GCN and GAT) on all four datasets, with and without the modified slope of ReLU (called Slope2GNN) for varying number of layers. Additionally, it presents results using SeLU, instead of ReLU, since it also modifies the slope of the function, while additionally normalizing node representations, as explained in section 2.4. To the best of our knowledge, this is the first time SeLU is used with GNNs.
Table 1 shows that methods with a modified activation function reduce oversmoothing significantly. As expected, baselines maintain a high performance for shallow architectures in homophilic datasets, where oversmoothing is not a problem. The modified activation functions (Slope2GNN and SeLU) consistently improve the testing accuracy as the number of layers ($\#L$) increases. Note that even the proposed method suffers performance degradation for very deep GNNs, due to the unavoidable nature of the oversmoothing (Oono & Suzuki, 2020). GCN and GAT average over information in the close neighborhood of the node, assuming homophily, which is not the case in the *Texas* dataset. Interestingly, low homophily in this dataset makes oversmoothing appear at a larger depth compared to other datasets. This is because connected nodes do not have similar representations and the model needs more Laplacian operations to oversmooth them.

**The effect of deeper networks on different activation functions:**
Focusing on the use of SeLU, although it seems to make the models resistant to oversmoothing, the effect seems to be lost for GCNs with 64 layers. This may be due to the sensitivity of the function on the choice of slope value, which is lower than 2. Moreover, the saturating region (part of SeLU used to reduce variance of data, if needed) of SeLU might further reduce the variance of node representations, acting in favor of oversmoothing. Finally, SNNs aim to keep the variance of the model stable, ignoring the effect of the Laplacian smoothing in GNNs.
To further examine the behavior of each method, as the number of layers increases, in Figure 1, we present a more detailed progression of the accuracy of GCNs, as the networks get deeper on the *Cora* dataset. The effect of oversmoothing and its reduction by the two methods that modify ReLU is very apparent here. Furthermore, the simpler Slope2GNN approach seems to be more resistant to the increase in the depth of the network.

**The need for deep networks:**
Having shown the benefit of using the modified activation functions in deep GNNs, we move to a set of experiments that aims to highlight also the value of using such deep architectures. In particular, we report our experimental results in the "cold start" scenario, as described in section 3.5. The cold-start datasets that we use in these experiments are generated by removing feature vectors

| # Layers | Method | Accuracy (%) | | | | | | | |
|---|---|---|---|---|---|---|---|---|---|
| | | Cora | | CiteSeer | | Pubmed | | Texas | |
| | | GCN | GAT | GCN | GAT | GCN | GAT | GCN | GAT |
| 2 | Original | 81.38 | 77.79 | 70.52 | **69.04** | 77.65 | **77.33** | **59.01** | **59.54** |
| | Slope2GNN | **81.84** | **77.81** | **70.55** | 68.24 | **78.34** | 77.02 | 57.93 | 56.67 |
| | SeLU | 81.74 | 76.59 | 69.91 | 66.81 | 77.80 | 76.83 | 59.01 | 57.21 |
| 4 | Original | 78.09 | 78.06 | 65.21 | 63.98 | **76.75** | **76.34** | **59.55** | 58.28 |
| | Slope2GNN | **80.39** | **79.54** | **67.57** | **67.27** | 76.46 | 75.59 | 57.66 | 57.39 |
| | SeLU | 79.85 | 79.33 | 67.53 | 67.24 | 74.09 | 72.62 | 58.02 | **58.65** |
| 8 | Original | 23.56 | 33.11 | 32.54 | 31.33 | 49.08 | 55.86 | **57.48** | **57.65** |
| | Slope2GNN | 76.54 | **78.33** | 65.80 | 65.55 | **75.00** | **74.20** | 57.18 | 56.85 |
| | SeLU | **79.41** | 78.19 | **67.27** | **67.58** | 73.43 | 71.93 | 56.85 | 57.47 |
| 16 | Original | 14.92 | 15.53 | 17.73 | 17.80 | 28.95 | 29.88 | 39.91 | 41.71 |
| | Slope2GNN | **76.81** | **75.00** | 57.10 | 56.13 | **76.35** | 73.81 | **57.21** | **57.53** |
| | SeLU | 76.38 | 74.92 | **61.99** | **62.23** | 76.11 | **74.27** | 52.88 | 57.44 |
| 32 | Original | 14.02 | 14.67 | 16.69 | 18.67 | 38.49 | 29.34 | 25.50 | 18.64 |
| | Slope2GNN | **73.21** | **69.62** | 47.20 | 49.16 | **75.78** | **74.63** | **55.79** | 54.14 |
| | SeLU | 68.77 | 69.36 | **47.90** | **52.97** | 71.80 | 73.60 | 55.77 | **57.75** |
| 64 | Original | 12.48 | 15.18 | 17.65 | 13.35 | 31.73 | 28.62 | 27.39 | 06.76 |
| | Slope2GNN | **61.59** | 24.17 | **46.46** | **26.24** | **71.68** | 38.01 | 46.31 | 41.44 |
| | SeLU | 26.85 | **30.55** | 27.10 | 20.42 | 40.90 | **41.00** | **58.38** | **56.30** |

Table 1: Performance comparison of vanilla GCN and GAT against SeLU and Slope2GNN enhanced versions of the same models, in *Cora, CiteSeer, Pubmed, Texas*. Average test node classification accuracy (%) for networks of different depth. With **bold** is the best performing model for each depth and each dataset.

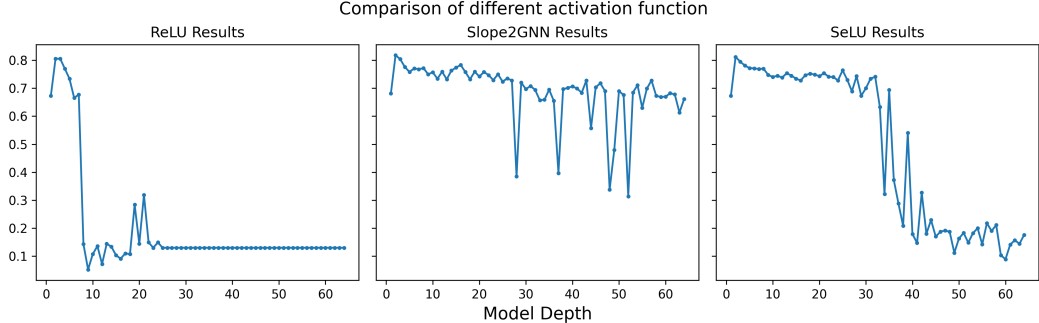

Figure 1: Comparison between three different activation functions for a GCN on *Cora* dataset. Y-axis shows model accuracy on test nodes, while varying the model's depth (shown in x-axis).

from unlabeled nodes and replacing them with all-zero vectors. For each combination of GNN and activation function, we present in Table 2 the best performance achieved and the depth at which the model attains that performance. The main observation in these results is the improvement in terms of accuracy with the use of deeper GNNs. This improvement is only attainable with the modified activation functions that reduce the effect of oversmoothing. The vanilla versions of GCN and GAT cannot go further than the performance of their shallow versions. Worth-mentioning is also the fact that the simple Slope2GNN approach manages to benefit from the deeper architectures even in the hard *Texas* dataset. This is not the case for SeLU, which may require tuning of the slope of the activation function, as discussed above.

## 5 RELATED WORK

Li et al. (2018) were the first to introduce the problem of oversmoothing in GNNs. An initial approach of activation function alternatives to avoid reduction of the rank of the feature space is the

| Model | | GCN | | GAT | |
|---|---|---|---|---|---|
| Dataset | Method | Accuracy (%) | #L | Accuracy (%) | #L |
| | Original | 64.85 | 4 | 59.74 | 3 |
| Cora | Slope2GNN | **73.47** | 19 | **72.78** | 20 |
| | SeLU | 73.00 | 22 | 72.65 | 23 |
| | Original | 41.94 | 4 | 38.50 | 4 |
| CiteSeer | Slope2GNN | **49.88** | 21 | **49.63** | 26 |
| | SeLU | 49.30 | 19 | 49.38 | 20 |
| | Original | 60.16 | 4 | 50.45 | 4 |
| Pubmed | Slope2GNN | **72.61** | 32 | 71.14 | 32 |
| | SeLU | 72.50 | 23 | **71.41** | 26 |
| | Original | 32.40 | 4 | 30.10 | 2 |
| Texas | Slope2GNN | **33.33** | 6 | **31.00** | 4 |
| | SeLU | 31.62 | 3 | 30.81 | 2 |

Table 2: Comparison of different models and activation functions on the "cold start" problem. We show accuracy percentage (%) for the test set using GCN and GAT as backbone GNN models. Only the features of the nodes in the training set are available to the model. We also show at what depth (i.e. # Layers) each model achieves its best accuracy.

one provided by Luan et al. (2019), where tanh is used instead of ReLU. Subsequent work, (Oono & Suzuki, 2020; Cai & Wang, 2020) further analyzed and theoretically proved the existence of the problem and showed that it is unavoidable as the depth increases. Xu et al. (2018) proposed Jumping Knowledge Networks (JK-Networks) as the first attempt to address the problem, by injecting skip connections to the GNNs. The model kept information from lower layers and combined them directly with the final layer, before labeling the node. Following a similar approach, Chen et al. (2020) proposed the use of residual connections between layers to enable deep architectures and alleviate the problem. Both of these approaches explicitly inject part of the initial information into higher layers of the network, implicitly downgrading the importance of intermediate layers. On the other hand, our method enables deep architectures without the need to inject initial information and allows the model to learn how to compress the initial information and what amount of it to maintain. We do not compare against these methods, because the aim of this work is to show that oversmoothing could be avoided without "short-circuiting" initial information to latter layers.

DropEdge (Rong et al., 2020) and DropNode (Do et al., 2021) are two alternative methods that address oversmoothing by altering the graph topology. In particular, they remove either edges or nodes at random, in order to slow down the message passing speed. In contrast to such approaches, our method does not change the graph topology, which may have unpredictable effects, but rather relies on a small adjustment of the slope of the activation function. An additional advantage of the approach proposed in this paper is that it does not have any hyper-parameters to be tuned, in contrast to all of the alternatives mentioned above. Hyperparameter tuning is also required by the Pairnorm method, which was mentioned in section 2.4.

## 6 CONCLUSION

In this paper we have shown the important role that the slope of the ReLU activation function plays in the oversmoothing problem in GNNs. We have further proposed a simple modification that reduces drastically the problem. We have illustrated the benefits of the approach in a set of experiments with different datasets and GNNs of different depth. Additionally, we showed the improvement in accuracy one can achieve through deep architectures that do not suffer from oversmoothing. This was evident in a set of experiments that simulated the "cold start" problem of missing node features in GNNs.

The simple link between the activation function and oversmoothing unveiled in this paper, opens up a range of interesting hypotheses to be investigated in the future. In particular, we would like to study the effect of changing the activation function in various GNN architectures and learning methods. Additionally, we would like to study alternative activation functions that could provide further benefits. Finally, we aim to study in more detail the approach of varying LR per layer.

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

APPENDIX

A: LEMMA 1 PROOF

**Lemma 1** *For a network of depth dep the total gradient reaching to the l-th layer $\left( i.e., \frac{dJ}{dw^{(l)}_{old_{i,j}}} \right)$ in order to update $W^{(l)}$ is bounded by:*

$$\frac{dJ}{dw^{(l)}_{old_{i,j}}} \leq \alpha^{(dep-l)} \cdot G_l$$

*Where J is the model's loss function (i.e. Cross Entropy), $\alpha$ stands for the ReLU slope and $G_l$ is the upper bound of gradients of the output of the subsequent layer (l+1) with respect to the weight elements of $W^{(l)}$.*

**Proof:** Firstly, let us define the gradient of the output of a ReLU function with respect to the input as follows:

$$\frac{d(ReLU(x))}{dx} = \begin{cases} \alpha, & x > 0 \\ 0, & x \leq 0 \end{cases}$$

Let us consider now a function of nested ReLUs, like the neural networks under investigation, i.e. $f(x) = g(\sigma(g(\sigma(..g(\sigma(g(x)))..)))))$, where $\sigma(\cdot)$ is the ReLU function applied repeatedly n times, and $g(\cdot)$ is a function that multiplies its input with a value. Then the gradient of $f(\cdot)$ with respect to x (x is a vector) is given as:

$$\frac{d(f(x))}{dx} = \begin{cases} \alpha^n \cdot \frac{d(g(x))}{dx}, & g(x) > 0, g(\sigma(\cdot)) > 0 \\ 0, & \text{otherwise} \end{cases} \tag{7}$$

This formula indicates the relation between slope, layer index, depth and largest singular value (we only care about the largest). Let as use the notion of J for the loss function of the network (i.e. Cross Entropy). The relation comes from the following formulas (for each weight element in the weight matrix):

$$w^{(l)}_{new_{i,j}} = w^{(l)}_{old_{i,j}} + \eta \cdot \frac{dJ}{dw^{(l)}_{old_{i,j}}}$$

$$\frac{dJ}{dw^{(l)}_{old_{i,j}}} = \alpha^{(depth-l)} \cdot \frac{dO^{(l+1)}}{w^{(l)}_{old_{i,j}}} \text{ or } 0$$

Where the 0 value comes from Equation 7. GNNs' forward pass is similar to $f(\cdot)$, hence performing the backpropagation leads to gradient calculation of the form of Equation 7. So the update rule adds (or subtracts) to each weight a big number (contains exponential factor). We prove, that $\frac{dJ}{dw_{old_{i,j}}}$ tends to be closer to zero as the layer index gets smaller, because when the gradient travels backwards the more distant it travels the more probable it is to die. There exist many components (gradients of ReLU functions) in the total gradient so it is probable one of them to be zero and the

total gradient to be zero.

Given Assumption 1, we get an upper bound regarding the gradient of the output of the subsequent layer with respect to each weight element of $W^{(l)}$ $\left( \text{i.e., } \frac{dO^{(l+1)}}{w^{(l)}_{old_{i,j}}} \leq G_l \right)$.

So we will have that:

$$\eta \cdot \frac{dJ}{dw_{old_{i,j}}} \leq \eta \cdot \alpha^{(depth-l)} \cdot G_l \leq \alpha^{(depth-l)} \cdot B_l \tag{8}$$

where $B_l$ is an upper bound on the product of the learning rate $\eta$ with $G_l$, i.e. $B_l = \eta \cdot G_l$.

B: LEMMA 2 PROOF

**Lemma 2** *While model's loss, through gradients, flows backwards some weight elements do not receive updates, because we have dying ReLUs (i.e. ReLUs outputting zero) (Lu et al., 2019). The total number of weight elements getting updated at layer l is bounded by:*

$$\#\{w^{(l)}_{i,j}\} \leq p^{(dep-l)} \cdot d^2$$

*where d is the largest of the two dimensions of $W^{(l)}$, i.e. there are at most $d^2$ elements in $W^{(l)}$, if it is a square matrix.*

**Proof:** Let us assume that the number of weights that get updated is reduced as the gradient flows backward. This is due to the fact that, the further the gradient 'travels' the more gradients of ReLUs will exist within it. Hence the greater is the probability at least one of them to be zero. In fact, we define the probability of a weight element to get updated and the number of weight elements to be updated in layer $l$ as:

$$P\{w^{(l)}_{i,j} = updated\} = p^{(depth-l)}$$

$$\#\{w^{(l)}_{i,j}\} = p^{(depth-l)} \cdot (i \cdot j) \leq p^{(depth-l)} \cdot d^2 \tag{9}$$

Where $i, j$ are the dimensions of the weight matrix and $d$ is the largest of them.

The probability for a weight element to get updated has the aforementioned form, because we have used the Assumption 2 regarding the probability of a ReLU unit not to output zero, which is at most $p$. Thus, the probability of getting updated is the probability of the gradient flowing backwards to reach the weight element, which in turn means not to have any ReLU component equal to zero.

The total number of elements that get updated is upper bounded (using union bound) on the sum of the probabilities of all elements of the weight matrix. In fact, every weight matrix has $i \cdot j$ elements where $i, j$ are the matrix dimensions and $d = max(i, j)$.

C: THEOREM 2 PROOF

**Theorem 2** *The upper bound of the largest singular value of the weight matrix $W^{(l)}$ at layer l for a GNN model, utilizing a ReLU activation function, depends on the slope of the function. That bound is given per layer and shows the effect of each iteration of updates on the weight matrix. We denote with $W_{old}$ and $W_{new}$ the weight matrices before and after the update during an iteration of the training process respectively.*

$$max(s_l(W^{(l)}_{new})) \leq ||W^{(l)}_{old}||_F + \sqrt{3} \cdot p^{\left(\frac{dep-l}{2}\right)} \cdot d \cdot \alpha^{(dep-l)} \cdot B_l$$

*where $B_l = \eta \cdot G_l$, dep is network's depth, d is the largest dimension of $W^{(l)}$ matrix, p is the upper bound of the probability of ReLU not to output zero and $\alpha$ is ReLU's slope.*

**Proof:** Regarding the singular values of a matrix (denoted by $s_l$), it is known that:

$$max(s_l(W^{(l)})) = ||W^{(l)}||_2 \leq ||W^{(l)}||_F = \sqrt{\sum |w^{(l)}_{i,j}|^2}$$

the matrix here being $W^{(l)}$, the weight matrix at layer $l$. Based on Lemma 1, a new weight, during the weight update process will be given by: $w_{new} \sim \alpha^{(depth-l)} \cdot B_l$. Since only $\#\{w_{i,j}^{(l)}\}$ weight elements are getting updated per layer (Lemma 2), in each iteration the Frobenius norm increases by a value, that depends on layer index, depth and slope. Specifically, if we define $\alpha^{(depth-l)} \cdot B_l = K_l$, in order to simplify the formulas, we have:

$$max(s_l(W_{new}^{(l)})) \leq ||W_{new}^{(l)}||_F = \sqrt{\sum |w_{new_{i,j}}^{(l)}|^2} \rightarrow$$

$$max(s_l(W_{new}^{(l)})) \leq \sqrt{\sum_{updated} |w_{new_{i,j}}^{(l)}|^2 + \sum_{not\_updated} |w_{new_{i,j}}^{(l)}|^2} \xrightarrow{8}$$

$$max(s_l(W_{new}^{(l)})) \leq \sqrt{\sum_{updated} |w_{old_{i,j}}^{(l)} + K_l|^2 + \sum_{not\_updated} |w_{old_{i,j}}^{(l)}|^2} \rightarrow$$

$$max(s_l(W_{new}^{(l)})) \leq \sqrt{\sum_{updated} \left(|w_{old_{i,j}}^{(l)}|^2 + |K_l|^2 + 2w_{old_{i,j}}^{(l)} K_l\right) + \sum_{not\_updated} |w_{old_{i,j}}^{(l)}|^2}$$

Using the fact that $w_{old_{i,j}}^{(l)} <= K_l$, because $K_l$ contains an exponential term and weights are initialized to values close to zero we get:

$$max(s_l(W_{new}^{(l)})) \leq \sqrt{\sum_{updated} \left(|w_{old_{i,j}}^{(l)}|^2 + 3|K_l|^2\right) + \sum_{not\_updated} |w_{old_{i,j}}^{(l)}|^2} \xrightarrow{\sqrt{A+B} \leq \sqrt{A} + \sqrt{B}}$$

$$max(s_l(W_{new}^{(l)})) \leq \sqrt{\sum_{updated} |w_{old_{i,j}}^{(l)}|^2 + \sum_{not\_updated} |w_{old_{i,j}}^{(l)}|^2} + \sqrt{\sum_{updated} 3|K_l|^2} \xrightarrow{Lemma2(9)}$$

In the second square root we sum over the updated weight elements and $K_l$ is independent of them. Hence, we get:

$$max(s_l(W_{new}^{(l)})) \leq ||W_{old}^{(l)}||_F + \sqrt{3|K_l|^2 \cdot \#\{w_{i,j}^{(l)}\}}$$

$$max(s_l(W_{new}^{(l)})) \leq ||W_{old}^{(l)}||_F + \sqrt{3} \cdot |K_l| \cdot p^{\left(\frac{depth-l}{2}\right)} \cdot d$$

$$max(s_l(W_{new}^{(l)})) \leq ||W_{old}^{(l)}||_F + \sqrt{3} \cdot p^{\left(\frac{depth-l}{2}\right)} \cdot d \cdot \alpha^{(depth-l)} \cdot B_l$$

So the slope of the function determines the upper bound of the Frobenius norm while training. In turn this norm is directly connected as an upper bound to the largest singular value of the matrix. In the aforementioned proof we have used previously defined Lemmas and Assumptions.

## D: MODIFYING LEARNING RATE: LR2GNN

As discussed in section 3.4, an alternative to changing the slope of ReLU is to modify the learning rate. In order to test this hypothesis and compare the results to the modified activation function, we performed some preliminary experiments with GCN. In these experiments, we determined a different learning rate per layer, using the validation part of each dataset and the chosen values

are shown in Table 3. We did this only for the first 8 layers, as the process of tuning the rate proved very cumbersome and time-consuming. The main observation seems to be the improved performance over the vanilla GCN for the three simpler datasets, coupled with a low accuracy for the *Texas* dataset. The comparison against the modified activation function seems inconclusive, but the latter approach wins due to its simplicity. If one opted for the modified learning rate, a method to automatically adapt the rate in different layers would be needed. Deeper architectures extend the complexity and the pool, from which we would have to find the proper values of learning rates. Worth to mention, combining different values of LR per layer with Slope2GNN results in negligible improvements.

| Layer index | 0 | 1 | 2 | 3 | 4 | 5 | 6 | 7 |
|---|---|---|---|---|---|---|---|---|
| Learning Rate | $3 \cdot 10^{-3}$ | $10^{-4}$ | $10^{-5}$ | $5 \cdot 10^{-5}$ | $10^{-5}$ | $10^{-4}$ | $10^{-4}$ | $10^{-4}$ |
| Cora Accuracy: **76.43** | | | CiteSeer Accuracy: **64.96** | | | Pubmed Accuracy: **78.82** | | |
| Texas Accuracy: **16.32** | | | | | | | | |

Table 3: Preliminary results of modifying the learning rate of GCN on *Cora, CiteSeer, Pubmed* and *Texas* datasets. The graph displays test accuracy of an 8-layer GCN, using the aforementioned learning rates.

E: DATASETS STATISTICS

| Datasets | # Nodes | # Edges | # Classes | # Features |
|---|---|---|---|---|
| Cora | 2708 | 5429 | 7 | 1433 |
| CiteSeer | 3327 | 4732 | 6 | 3703 |
| Pubmed | 19717 | 44338 | 3 | 500 |
| Texas | 183 | 309 | 5 | 1703 |

Table 4: The statistics of all datasets used in this work.

F: EXTENDED EXPERIMENTS

**Extended experimentation, regular datasets:**
Table 5 is an expanded version of Table 1, showing average test node classification accuracy along with the respective standard deviations. We have also included GCNII (Chen et al., 2020), which is one of the best performing GNNs utilizing residual connections. GCNII reduces oversmoothing through 'short circuiting' initial information to subsequent layers of the model. The aim of the comparison against GCNII is to show that by changing the slope of ReLU simple models are capable of producing comparable (up to a limit) results to more sophisticated architectures, that use residual connections.

| Accuracy (%) and standard deviation | | | | | | | |
|---|---|---|---|---|---|---|---|
| Dataset | Method | # Layers | | | | | |
| | | 2 | 4 | 8 | 16 | 32 | 64 |
| Cora | GCN | $81.38_{\pm0.5}$ | $78.09_{\pm1.8}$ | $23.56_{\pm9.2}$ | $14.92_{\pm6.4}$ | $14.02_{\pm3.5}$ | $12.48_{\pm2.9}$ |
| | GCN(Slope2GNN) | $\mathbf{81.84}_{\pm0.3}$ | $\mathbf{80.39}_{\pm0.9}$ | $76.54_{\pm1.5}$ | $\mathbf{76.81}_{\pm2.2}$ | $\mathbf{73.21}_{\pm1.7}$ | $\mathbf{61.59}_{\pm4.7}$ |
| | GCN(SELU) | $81.74_{\pm0.4}$ | $79.85_{\pm0.7}$ | $\mathbf{79.41}_{\pm0.5}$ | $76.38_{\pm1.1}$ | $68.77_{\pm5.7}$ | $26.85_{\pm4.0}$ |
| | GAT | $77.79_{\pm1.5}$ | $78.06_{\pm1.6}$ | $33.11_{\pm9.2}$ | $15.53_{\pm9.0}$ | $14.67_{\pm4.5}$ | $15.18_{\pm5.5}$ |
| | GAT(Slope2GNN) | $\mathbf{77.81}_{\pm2.2}$ | $\mathbf{79.54}_{\pm1.2}$ | $\mathbf{78.33}_{\pm2.5}$ | $\mathbf{75.00}_{\pm1.9}$ | $\mathbf{69.62}_{\pm2.4}$ | $24.17_{\pm9.5}$ |
| | GAT(SELU) | $76.59_{\pm2.2}$ | $79.33_{\pm1.1}$ | $78.19_{\pm1.1}$ | $74.92_{\pm1.6}$ | $69.36_{\pm4.3}$ | $\mathbf{30.55}_{\pm4.9}$ |
| | GCNII | $\mathbf{82.32}_{\pm0.6}$ | $\mathbf{82.89}_{\pm0.6}$ | $\mathbf{83.99}_{\pm0.5}$ | $\mathbf{84.77}_{\pm0.7}$ | $\mathbf{84.96}_{\pm0.7}$ | $\mathbf{85.39}_{\pm0.6}$ |
| CiteSeer | GCN | $70.52_{\pm0.6}$ | $65.21_{\pm1.0}$ | $32.54_{\pm9.3}$ | $17.73_{\pm4.9}$ | $16.69_{\pm4.8}$ | $17.65_{\pm4.2}$ |
| | GCN(Slope2GNN) | $\mathbf{70.55}_{\pm0.4}$ | $\mathbf{67.57}_{\pm0.7}$ | $65.80_{\pm1.2}$ | $57.10_{\pm4.3}$ | $47.20_{\pm3.8}$ | $\mathbf{46.46}_{\pm4.2}$ |
| | GCN(SELU) | $69.91_{\pm0.4}$ | $67.53_{\pm0.6}$ | $\mathbf{67.27}_{\pm0.9}$ | $\mathbf{61.99}_{\pm1.9}$ | $\mathbf{47.90}_{\pm9.2}$ | $27.10_{\pm7.6}$ |
| | GAT | $\mathbf{69.04}_{\pm1.4}$ | $63.98_{\pm2.8}$ | $31.33_{\pm7.0}$ | $17.80_{\pm1.9}$ | $18.67_{\pm2.8}$ | $13.35_{\pm4.3}$ |
| | GAT(Slope2GNN) | $68.24_{\pm1.2}$ | $\mathbf{67.27}_{\pm1.0}$ | $65.55_{\pm2.0}$ | $56.13_{\pm2.5}$ | $49.16_{\pm4.5}$ | $\mathbf{26.24}_{\pm3.5}$ |
| | GAT(SELU) | $66.81_{\pm1.5}$ | $67.24_{\pm1.1}$ | $\mathbf{67.58}_{\pm1.1}$ | $\mathbf{62.23}_{\pm3.8}$ | $\mathbf{52.97}_{\pm6.4}$ | $20.42_{\pm2.8}$ |
| | GCNII | $\mathbf{68.05}_{\pm0.7}$ | $67.75_{\pm1.3}$ | $\mathbf{70.90}_{\pm0.5}$ | $\mathbf{72.69}_{\pm0.4}$ | $72.68_{\pm0.4}$ | $\mathbf{72.77}_{\pm0.9}$ |
| Pubmed | GCN | $77.65_{\pm0.3}$ | $\mathbf{76.75}_{\pm0.7}$ | $49.08_{\pm9.7}$ | $28.95_{\pm9.9}$ | $38.49_{\pm9.8}$ | $31.73_{\pm9.9}$ |
| | GCN(Slope2GNN) | $\mathbf{78.34}_{\pm0.1}$ | $76.46_{\pm1.0}$ | $75.00_{\pm1.4}$ | $\mathbf{76.35}_{\pm1.7}$ | $\mathbf{75.78}_{\pm2.1}$ | $\mathbf{71.68}_{\pm1.7}$ |
| | GCN(SELU) | $77.80_{\pm0.2}$ | $74.09_{\pm1.3}$ | $73.43_{\pm1.1}$ | $76.11_{\pm2.1}$ | $71.80_{\pm4.3}$ | $40.90_{\pm0.9}$ |
| | GAT | $\mathbf{77.33}_{\pm0.7}$ | $\mathbf{76.34}_{\pm1.3}$ | $55.86_{\pm7.5}$ | $29.88_{\pm9.9}$ | $29.34_{\pm9.9}$ | $28.62_{\pm9.9}$ |
| | GAT(Slope2GNN) | $77.02_{\pm0.7}$ | $75.59_{\pm1.1}$ | $\mathbf{74.20}_{\pm1.7}$ | $73.81_{\pm2.4}$ | $\mathbf{74.63}_{\pm1.1}$ | $38.01_{\pm5.7}$ |
| | GAT(SELU) | $76.83_{\pm1.0}$ | $72.62_{\pm1.2}$ | $71.93_{\pm1.8}$ | $\mathbf{74.27}_{\pm2.7}$ | $73.60_{\pm2.2}$ | $\mathbf{41.00}_{\pm0.3}$ |
| | GCNII | $\mathbf{78.57}_{\pm0.6}$ | $\mathbf{79.09}_{\pm0.4}$ | $\mathbf{79.85}_{\pm0.5}$ | $79.70_{\pm0.3}$ | $79.77_{\pm0.2}$ | $79.65_{\pm0.3}$ |
| Texas | GCN | $\mathbf{59.01}_{\pm2.7}$ | $\mathbf{59.55}_{\pm7.7}$ | $\mathbf{57.48}_{\pm6.5}$ | $39.91_{\pm0.7}$ | $25.50_{\pm0.6}$ | $27.39_{\pm0.9}$ |
| | GCN(Slope2GNN) | $57.93_{\pm4.2}$ | $57.66_{\pm5.0}$ | $57.18_{\pm6.2}$ | $\mathbf{57.21}_{\pm6.4}$ | $\mathbf{55.79}_{\pm6.5}$ | $46.31_{\pm4.1}$ |
| | GCN(SELU) | $59.01_{\pm2.7}$ | $58.02_{\pm5.0}$ | $56.85_{\pm6.2}$ | $52.88_{\pm6.6}$ | $55.77_{\pm6.2}$ | $\mathbf{58.38}_{\pm6.8}$ |
| | GAT | $\mathbf{59.54}_{\pm4.2}$ | $58.28_{\pm6.5}$ | $\mathbf{57.65}_{\pm6.7}$ | $41.71_{\pm4.0}$ | $18.64_{\pm2.9}$ | $06.76_{\pm1.9}$ |
| | GAT(Slope2GNN) | $56.67_{\pm4.3}$ | $57.39_{\pm5.2}$ | $56.85_{\pm6.0}$ | $\mathbf{57.53}_{\pm6.1}$ | $54.14_{\pm6.5}$ | $41.44_{\pm2.3}$ |
| | GAT(SELU) | $57.21_{\pm5.0}$ | $\mathbf{58.65}_{\pm5.8}$ | $57.47_{\pm4.5}$ | $57.44_{\pm6.9}$ | $\mathbf{57.75}_{\pm6.4}$ | $\mathbf{56.30}_{\pm5.1}$ |
| | GCNII | $\mathbf{69.37}_{\pm2.9}$ | $\mathbf{70.72}_{\pm4.3}$ | $\mathbf{72.07}_{\pm2.5}$ | $70.00_{\pm2.2}$ | $\mathbf{71.26}_{\pm1.6}$ | $\mathbf{71.35}_{\pm1.1}$ |

Table 5: Performance comparison of vanilla GCN and GAT against SeLU and Slope2GNN enhanced versions of the same models, in *Cora, CiteSeer, Pubmed, Texas*. We include GCNII (Chen et al., 2020) in the comparison. Average test node classification accuracy (%) and standard deviation for networks of different depth. With **bold** the best performing method for each model (i.e., GCN, GAT and GCNII), depth and dataset.

**Extended "Cold start":**
Table 6 is an expanded version of Table 2 including GCNII's performance and the standard deviations for all models. The "cold start" problem requires deeper models, because nodes containing informative feature vectors are located further from each test node. Thus, architectures that utilize deep GNNs could effectively address the problem, if they avoid oversmoothing. Once again our method effectively enables deep architectures using simple models (i.e., GCN, GAT).

| Dataset | Method | Accuracy (%) & std | #L |
|---|---|---|---|
| Cora | GCN | $64.85_{\pm1.2}$ | 4 |
| | GCN(Slope2GNN) | $\mathbf{73.47}_{\pm0.9}$ | 19 |
| | GCN(SeLU) | $73.00_{\pm1.5}$ | 22 |
| | GAT | $59.74_{\pm1.2}$ | 3 |
| | GAT(Slope2GNN) | $\mathbf{72.78}_{\pm1.4}$ | 20 |
| | GAT(SeLU) | $72.65_{\pm1.6}$ | 23 |
| | GCNII | $\mathbf{73.40}_{\pm1.0}$ | 5 |
| CiteSeer | GCN | $41.94_{\pm0.3}$ | 4 |
| | GCN(Slope2GNN) | $\mathbf{49.88}_{\pm1.1}$ | 21 |
| | GCN(SeLU) | $49.30_{\pm1.2}$ | 19 |
| | GAT | $38.50_{\pm4.7}$ | 4 |
| | GAT(Slope2GNN) | $\mathbf{49.63}_{\pm0.9}$ | 26 |
| | GAT(SeLU) | $49.38_{\pm1.3}$ | 20 |
| | GCNII | $\mathbf{59.84}_{\pm3.0}$ | 31 |
| Pubmed | GCN | $60.16_{\pm5.4}$ | 4 |
| | GCN(Slope2GNN) | $\mathbf{72.61}_{\pm0.8}$ | 32 |
| | GCN(SeLU) | $72.50_{\pm0.8}$ | 23 |
| | GAT | $50.45_{\pm8.9}$ | 4 |
| | GAT(Slope2GNN) | $71.14_{\pm1.5}$ | 32 |
| | GAT(SeLU) | $\mathbf{71.41}_{\pm1.7}$ | 26 |
| | GCNII | $\mathbf{78.25}_{\pm0.6}$ | 7 |
| Texas | GCN | $32.40_{\pm6.8}$ | 4 |
| | GCN(Slope2GNN) | $\mathbf{33.33}_{\pm6.9}$ | 6 |
| | GCN(SeLU) | $31.62_{\pm5.9}$ | 3 |
| | GAT | $30.10_{\pm8.5}$ | 2 |
| | GAT(Slope2GNN) | $\mathbf{31.00}_{\pm5.9}$ | 4 |
| | GAT(SeLU) | $30.81_{\pm5.8}$ | 2 |
| | GCNII | $\mathbf{57.66}_{\pm0.0}$ | 1 |

Table 6: Comparison of different models and activation functions on the "cold start" problem. We show accuracy percentage (%) and standard deviation on the test set using GCN and GAT as backbone GNN models and GCNII as a representative of residual GNNs. Only the features of the nodes in the training set are available to the model. We also show at what depth (i.e. # Layers) each model achieves its best accuracy.

**Slope sensitivity results:**
The choice of slope (i.e., $\alpha = 2$) that we proposed is based on Equation 6. We have verified experimentally that this slope value helps to reduce the effect of oversmoothing. However, this is not the only slope value that could be used. Therefore, we experimented with other slope values and present the results in Figure 2. Based on the Figure, it is evident that any value between 1.8 and 2.2 reduces oversmoothing, making the exact choice of value less important.

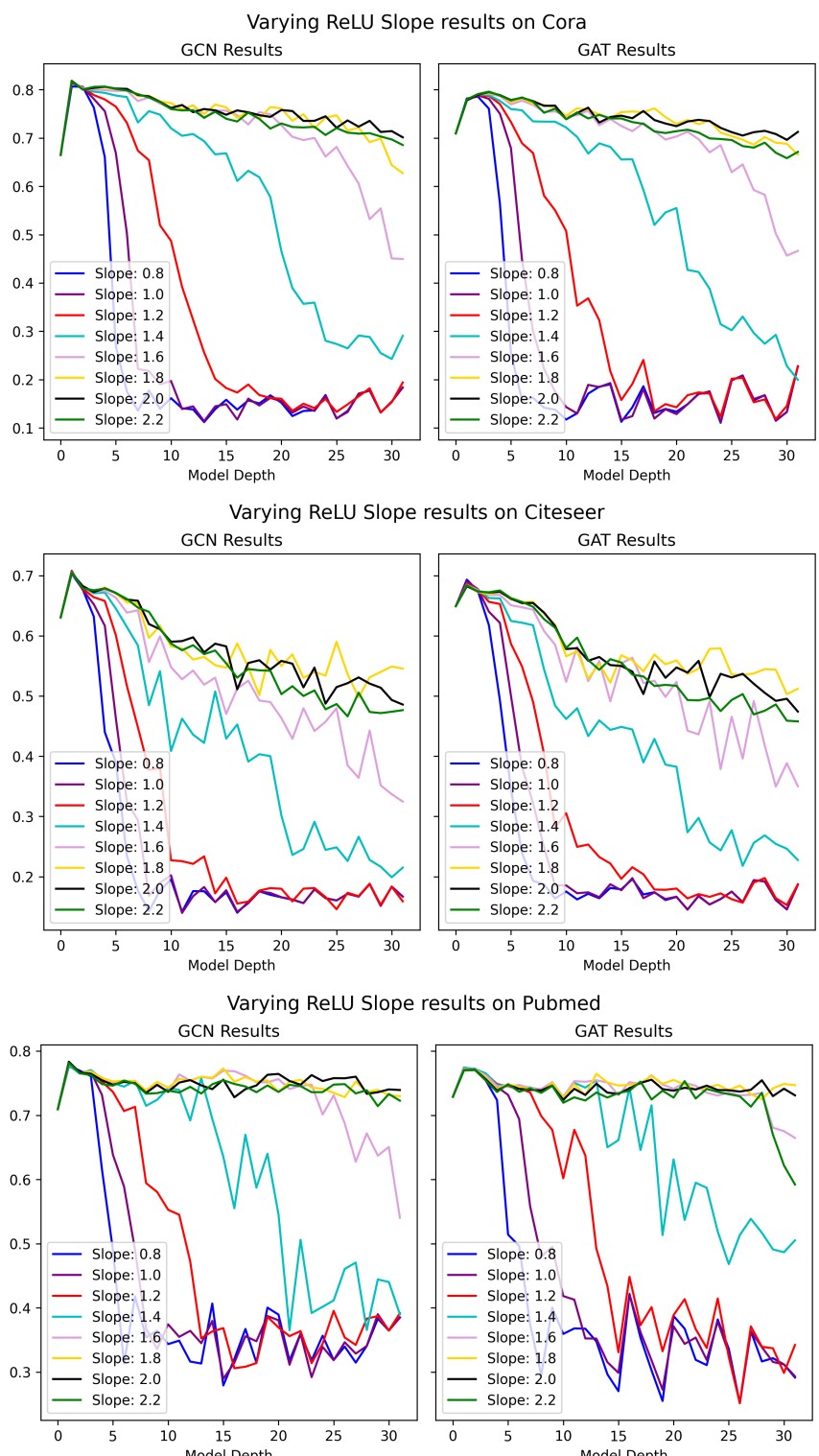

Figure 2: Comparison of test performance accuracy (%) of GCN and GAT with varying slope values in *Cora, CiteSeer* and *Pubmed*. We show the accuracy of the models in y-axis, while increasing their depth.

**Visualising the effect of oversmoothing:**
In order to understand the effect of oversmoothing in GNNs and show how our method alleviates it, we visualise node representations using t-SNE (van der Maaten & Hinton, 2008) for varying depth models. Figures 3, 4 and 5 visualise the representations for different depth levels and different ReLU slope values. Ideally, we would like nodes of the same color (same class) to be close together and well-separated from other classes. In all three datasets, we observe how oversmoothing (at slope = 1) leads to a mixing of the classes. We also see how the situation changes for slope = 2. In those graphs, node representations do not rapidly mix as the depth of the network increases.

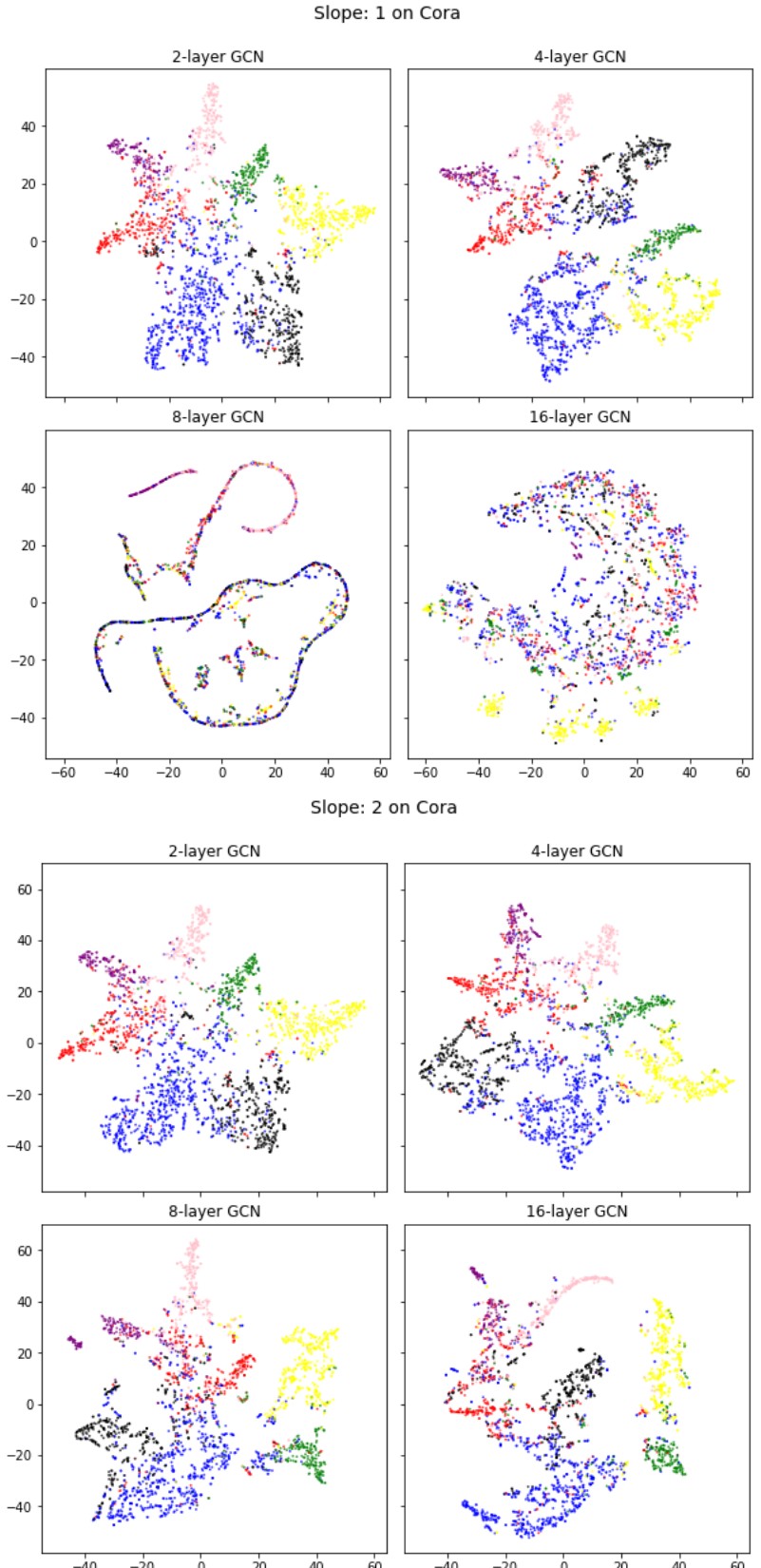

Figure 3: t-SNE of node representations of GCN and GAT on *Cora*, while increasing model's depth. Upper (lower) 4 figures show the results without (with) our method.

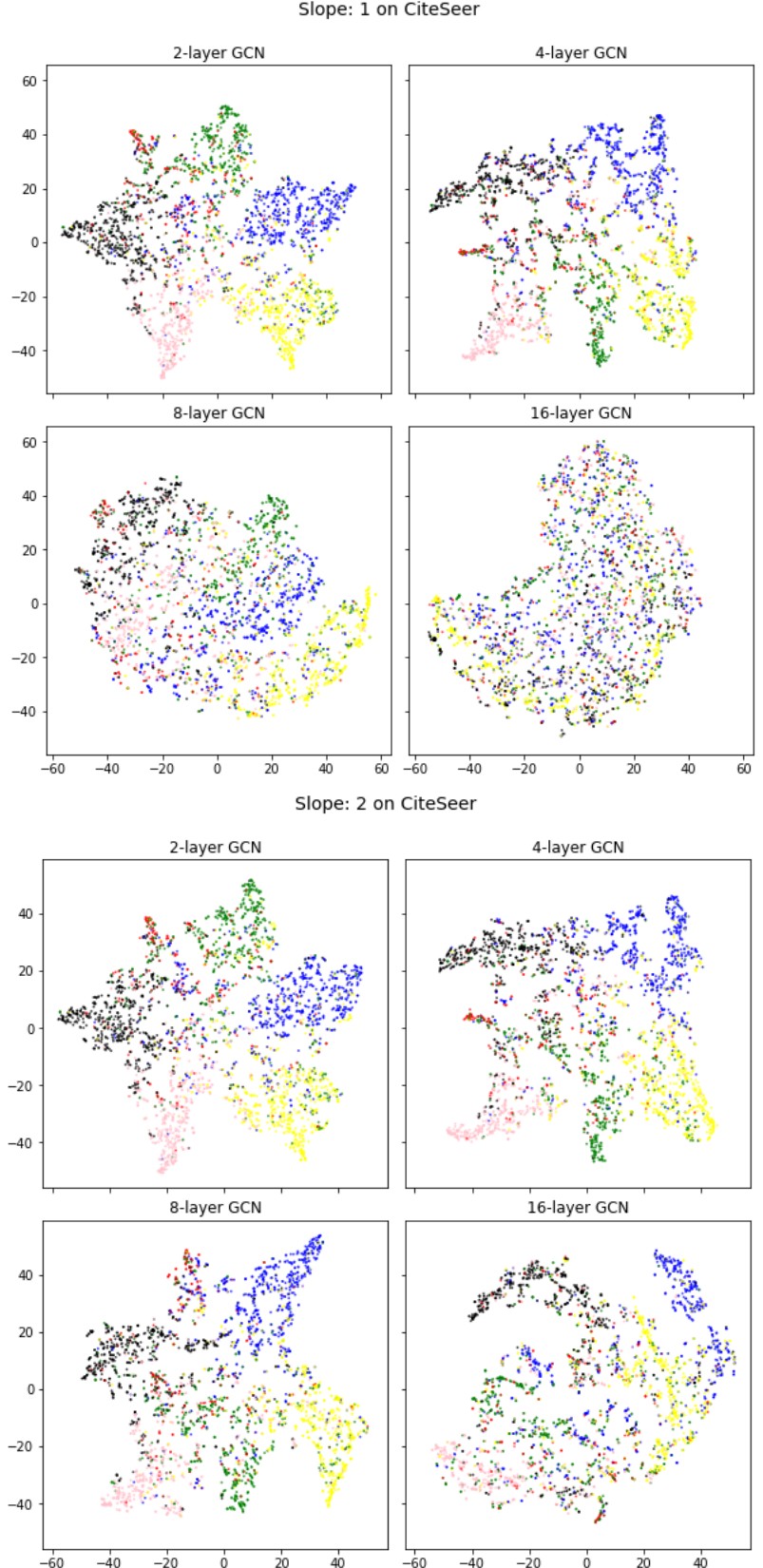

Figure 4: t-SNE of node representations of GCN and GAT on *CiteSeer*, while increasing model's depth. Upper (lower) 4 figures show the results without (with) our method.

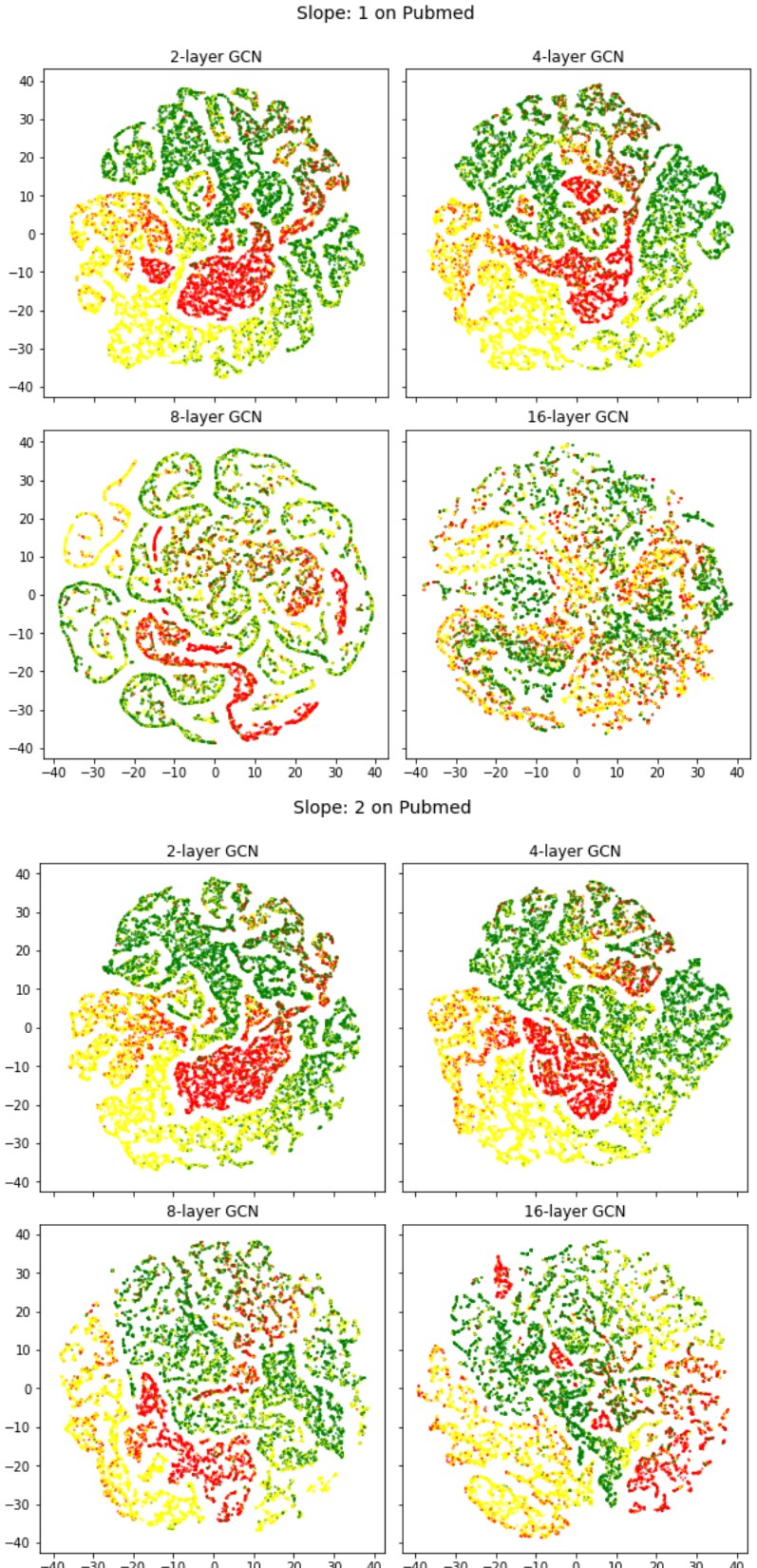

Figure 5: t-SNE of node representations of GCN and GAT on *Pubmed*, while increasing model's depth. Upper (lower) 4 figures show the results without (with) our method.

