# OpenReview forum: "REDUCING OVERSMOOTHING IN GRAPH NEURAL NETWORKS BY CHANGING THE ACTIVATION FUNCTION"
_ICLR.cc/2023/Conference — Submitted to ICLR 2023_

### Official Review · Reviewer_XeNT · 2022-10-22

**Confidence:** 4
**Correctness:** 3
**Technical Novelty And Significance:** 2
**Empirical Novelty And Significance:** 2
**Recommendation:** 5

**Clarity, Quality, Novelty And Reproducibility:**

Though the paper is sufficiently well written, the theoretical claims (in Lemmas and Theorems) are given without the necessary rigor.

**Strength And Weaknesses:**

Strenghts:
- The paper highlights the relevant role of the slope of the ReLU activation function in the design of deep GNN, showing the potentialities of increasing the value of the slope as a simple way to counter-balance the oversmoothing effects.

Weaknesses:
- The mathematical analysis lacks some rigor in the determination of the exact value of the slope.
In Section 3, which represents the core of this contribution, the paper elaborates on the importance of the value of the ReLU slope (i.e., alpha) on the upper bound for the largest singular value of the weight matrix. In a rather vague way, and without a formal statement it is concluded that the right value is alpha = 2. While this is a consequence of intuitive reasoning (and supported by some vague sentences, like "a slope of 2 makes the second exponential factor prevail over the first one", without a reference to the equation), I could not see a mathematical argument for which alpha needs to be exactly 2. Again, in a non-rigorous way it is stated that "we cannot make the slope too large", and "if we increase the slope of ReLU too much, the upper bound in eq. 6 becomes too loose". How much is too much? Is there any mathematically grounded approach to show that any larger (or smaller) value than alpha = 2 leads to gradient problems? The main message I got from the paper is that the actual value of alpha would be better tuned instead of being fixed to 1. This could be probably be treated as a hyperparameter and chosen accordingly.
- Hyperparameters tuning is essentially not performed in the experiments. Indeed, in Section 4.1 it is reported that the crucial values of the hyperparameters are all fixed (also, I could not find information on the used learning rate). Moreover, it could be interesting to tune the value of alpha as a hyperparameter.
- Results in the experimental section are reported without std across repetitions, and without any analysis of statistical significance of the reported average accuracy.

**Summary Of The Paper:**

The paper studies the relevance of the slope of the ReLU activation function in deep GNNs with respect to the over smoothing issue.
The proposed strategy is simply to use a higher slope for the ReLU, setting its value to 2.
The paper presents an experimental analysis in which this idea is applied to GCN and GAT on a number of benchmark problems, and a modified version called cold start (in which the input features are removed from some nodes).

Update after rebuttal: I thank the authors for considering my concerns on their work. The rebuttal partially clarified doubts regarding the validity (especially) of the experimental analysis (and lack of fine-tuning). I am happy to slightly raise my score accordingly.

**Summary Of The Review:**

The paper has the merit of highlighting the role of the ReLU slope on the over smoothing problem. However, in its current form there are several issues with the mathematical analysis (and presentation), as well as with the experimental section.

---

> ### Author Response · Authors · 2022-11-15
> **Response to Reviewer XeNT**
>
> Thank you for the comments and helpful feedback on our work. **We want to inform the reviewers that the manuscript has been updated including an extended experimental subsection, Appendix F.** The aim of the paper is to show that a simple method, applicable to many architectures can reduce the problem of oversmoothing.
>
> **Mathematical rigor.** We agree with the reviewer that the choice of slope=2 is not exact, but it is one that agrees with our theoretical analysis. In fact, the exact choice of slope value is not important and we consider this to be an advantage of the proposed method. To illustrate this point, we have included in the new Appendix F experimental results for different slope values as depth increases.
>
> **Hyper-parameter tuning.** The parameters of the NN have not been extensively fine-tuned, similar to related work, such as Pairnorm (Zhao & Akoglu, 2020), GCNII (Chen et. al, 2020), and FastGCN (Chen et. al, 2018). The learning rate was indeed fine-tuned and the value $10^{-3}$ was chosen (see new sentence in section 4.1, hyperparameters paragraph in the updated manuscript version). Furthermore, we agree on the need to present the results for different slope values and have included in Appendix F the results for different slope values and varying depth for GCN and GAT. Our results show that the proposed method is not sensitive to the exact choice of slope value, as long as it is close to 2.
>
> **Results with standard deviations.** We agree with the reviewer about the need to include such information and have incorporated it in the extended versions of Table 1 and Table 2 in Appendix F.

---

### Official Review · Reviewer_kwRc · 2022-10-24

**Confidence:** 4
**Correctness:** 3
**Technical Novelty And Significance:** 2
**Empirical Novelty And Significance:** 2
**Recommendation:** 3

**Clarity, Quality, Novelty And Reproducibility:**

- Clarity: The paper is well-written, and the method is clear, surprisingly simple, and well-established.

- Quality: The paper is of decent quality. However, the simplicity of the method is a bit underwhelming. That said, additional experimentation discussed above would definitely increase its quality.

- Novelty: Though simple, the adaptation of the activation function seems to be quite novel, especially in its application to mitigate over-smoothing.

- Reproducibility: The authors do not provide code. That said, the method is quite simple to replicate even with no code and training configurations provided in the main paper. However, it would help to have code to reproduce the strongest results of the “cold start” experiments.


**Strength And Weaknesses:**

Strengths
- The paper is well presented. The theorems are clear, and the theoretical foundation for the proposed method is well established.
- Current experiments provide some support for the hypothesis that the method can mitigate over-smoothing in graph neural networks

Weaknesses
- Just a comment: The phenomenon of over-smoothing is a case where the graph feature maps become too smooth so that they lose the information necessary to make a prediction. It is not necessarily the phenomenon where the accuracy degrades as we add more layers. The latter can be caused simply by a failure to optimize, and not caused by a flawed architecture.

- The claims in this paper are too bold. Table 1 clearly demonstrates that over-smoothing still exists in the proposed method.

- There are multiple new networks that do not suffer from over-smoothing. E.g., GCNII (2020), GRAND (Chamberlain et al., 2021), PDE-GCN (Eliasof et al. 2021), GraphCon (Rusch et al, 2022), EGNN (Zhou 2021). I am not sure that tackling the over-smoothing problem is still a pressing research question in this field. GCN and GAT are indeed over-smooth, but they are relatively old by now.

- The authors do not compare their method to other GNNs.
- I believe that the authors do not provide sufficient investigations into the sensitivity and generalization of the proposed solution. Specifically, I believe several experiments are missing:

1. Generalization to modern architectures and activation functions: current
architectures and activation functions explored in this paper are from no later than 2019. Though the authors explicitly state that they intentionally do not compare to
modern architectures which use skip connections, I do not agree that these methods
are out of the scope of this paper. This is because the proposed method has the
potential to provide benefits to a vast variety of graph neural networks in overcoming
over-smoothing and I believe it is important to show that the method generalizes to
other more modern architectures.
2. Sensitivity: I believe additional experimentation is required to support the choice of
the relu factor of 2 in this paper, such as trying different values of this factor. For
example, if I understand correctly, eq. (6) suggests that a factor of sqrt(2) may be more appropriate in
counteracting the exponential decay of the upper bound.


**Summary Of The Paper:**

The authors build upon the recent analysis in [1], which provides a theoretical foundation for the over-smoothing phenomenon in graph neural networks, to propose an adaptation of the activation function to counteract this effect which harms performance. Specifically, they propose multiplying the relu by a factor of 2, to counteract an exponential decay factor of 0.5, which contributes to over-smoothing. The authors provide both theoretical and empirical evidence that their proposed method indeed mitigates over-smoothing, thus enabling deeper graph neural networks to be used without significant performance drops.

**Summary Of The Review:**

The paper is well written, the problem statement well established, and the proposed solution well founded. However, it is clear that the method is insufficient to mitigate over-smoothing, and at this point, many new architectures do not suffer from it. I also think that the authors do not perform sufficient investigations into the generalizability of their proposed method to modern architectures and activation functions (later than 2018), nor do they provide in-depth investigations into the sensitivity of the proposed solution to different choices of relu factors.

---

> ### Author Response · Authors · 2022-11-15
> **Response to Reviewer kwRc**
>
> Thank you for the comments and helpful feedback on our work. **We want to inform the reviewers that the manuscript has been updated including an extended experimental subsection, Appendix F.** The aim of the paper is to show that a simple method, applicable to many architectures can reduce the problem of oversmoothing.
>
> **Eliminating oversmoothing.** Suzuki & Oono (2020) have shown that oversmoothing is unavoidable. Our method reduces the effect of oversmoothing in deep GNNs, but does not eliminate it. To further illustrate the effect of the proposed method, we have included (Appendix F) t-SNE (van der Maaten and Hinton, 2008) results of GCN and GAT with and without the modified activation function.
>
> **Is oversmoothing still a problem?** Recent literature, some of which was suggested by the reviewer, indicates that the problem of oversmoothing has not been fully solved yet for GNNs. The simple method proposed in our paper addresses the problem to a large extent, avoiding complex architectures.
>
> **Comparison with other GNNs.** The basic claim of our work is that methods that suffer from the problem of oversmoothing can be improved significantly in a very simple way. Residual architectures deal with oversmoothing in a more complex manner.  For reasons of completeness, we have extended our experiments to compare our results against residual architectures. The updated manuscript includes results of GCNII (Chen et. al, 2020) in Appendix F, which extend those in Table 1 and Table 2.
>
> **Generalization to modern architectures and activation functions.** The proposed method is applicable to architectures that rely heavily on ReLU, as an activation function. We wanted to show that even simple architectures can be improved significantly with a small modification of the function. Many of the most recent methods that deal with oversmoothing, e.g. using residual connections, lower significantly their dependence on the activation function and its slope. Our method is less useful for such methods.
>
> **Slope sensitivity.** We agree with the reviewer regarding the need to show the results for different slope values and we have included them in Appendix F. The results confirm the choice of slope=2 and the method does not seem to be sensitive to the exact value that is chosen. Moreover, to better illustrate the effect of changing the slope to the representations of the nodes, we have included results of t-SNE (van der Maaten and Hinton, 2008) with slope 1 and slope 2 in Appendix F.

---

> > ### Comment · Reviewer_kwRc · 2022-11-21
> > **Response to rebuttal**
> >
> > Dear Authors,
> > Thank you for your detailed rebuttal. I still have several of the concerns that I wrote in my review. It is true that at the time, the methods that I mentioned stated that over-smoothing is still a problem, but since 5 of them introduce architectures that do not over-smooth (and there are more by now, from recent conferences such as ICML and NeurIPS 2022), it cannot be such a pressing problem. How can over-smoothing be unavoidable when 5 architectures do not over-smooth? That is a contradiction that the authors may target. Moreover, the method the authors suggest does not solve the problem. The new variants of GCN and GAT still over-smooth, as is demonstrated in the new results that the authors added. Even if some variants do work here and there, every variant is good for one data set and severely over-smooth for another. I, therefore, keep my original score.
> > Best regards,
> > Reviewer kwRc

---

> > > ### Author Response · Authors · 2022-11-25
> > > **Thank you for the comments**
> > >
> > > Thank you for the comments on the updated manuscript.
> > >
> > > It is true that a number of recent methods claim to have solved the problem. However, deeper analysis of these methods and experimentation with other datasets is needed to confirm that this is indeed the case. In fact, there are other recent publications, which also consider the problem open. For example (Huang et. al, ICLR 2022) refer to methods based on residual connections, but attempt to solve the problem differently. Furthermore, in (Cong et. al, NIPS 2021) a simple GCN seems to outperform in some cases the residual-based methods. If it turns out that methods based on residual connections only partially address the problem of oversmoothing, we may need to revisit basic GNN methods. At that stage, knowing about the role of the slope of ReLU and how it can be used to reduce the problem will prove very valuable. It is for this reason that we find the results of our work worth-publishing at ICLR 2023.
> > >
> > > We hope we have addressed all of the other  concerns.

---

> > > > ### Comment · Reviewer_kwRc · 2022-11-25
> > > > **discussioin**
> > > >
> > > > I still do not fully agree, but I'll appreciate it if you send me the full names of the papers you mention to see if there's still an issue of over-smoothing. I could not find them in your paper. Also, if they are relevant to motivate the discussion, as you say, you should cite them and discuss their contributions in your paper. By now, I know quite a lot of methods that do not over-smooth (some I wrote, and some newer). If a theory says otherwise, I'll be happy to learn about it and examine it.
> > > >
> > > > This does not change the fact that your proposed method does not solve the problem, it only mitigates it to some extent.

---

> > > > > ### Author Response · Authors · 2022-11-25
> > > > > **Discussion**
> > > > >
> > > > > “On Provable Benefits of Depth in Training Graph Convolutional Networks” (Cong et. al, NIPS 2021), see Table 2
> > > > >
> > > > > C-DropEdge (TOWARDS DEEPENING GRAPH NEURAL NETWORKS: A GNTK-BASED OPTIMIZATION PERSPECTIVE, Huang et. al, ICLR 2022) and DropEdge (Rong et. al, ICLR 2020)

---

### Official Review · Reviewer_rfT6 · 2022-10-25

**Confidence:** 3
**Correctness:** 3
**Technical Novelty And Significance:** 3
**Empirical Novelty And Significance:** 3
**Recommendation:** 5

**Clarity, Quality, Novelty And Reproducibility:**

The paper is well written and easy to follow. There are some assumptions that are made through the paper that it is not clear why it holds for GNNs. For example the second assumption which holds for FFs is used for GNNs but I think GNNs are more restricted than FFNs because of the local aggregation and the probability that Relu(h)!=0 should be greater that Relu(h)!=0 in FFs.


**Strength And Weaknesses:**

Strength:
1- The paper developed a simple approach that reduces oversmoothing and enables deep architectures. It is known that in deep GNNs repetition of the Laplacian operator acts as a variance reducer and causes the oversmoothing. The paper transfers the idea of SeLU to GNNs, focusing on the part that increases the variance to alleviate oversmoothing.

2- The paper provides a theoretical analysis that the convergence to a subspace where node representations are the same can be controlled by two factors 1) slope of the Relu function and 2) modifying the learning rate.

3- Since determining the right LR per layer is a non-trivial task, they focused on the slope of the Relu function. Increasing the slope of ReLU too much also causes exploding gradients and their analysis of the upper bound of the largest singular value of the weight matrix becomes too loose.

Weakness:

1- Even though the paper provides an interesting analysis that correlates the slope of the Relu and the learning rate with the largest singular value of the weight matrix which affects the oversmoothing phenomena, it is still not clear how it is useful in the real scenario given the fact that even their experimental results show that still shallow GNNs performs better in classification task.

2- The paper claims that deep GNNs are beneficial in the presence of cold-start problems, but their experimental results sound artificial given that the features of the test data are removed. The cold start problem may appear on feature level (some features are missing) or edge level (some connections are not formed yet). So to show the impact of the deep GNNs I would suggest running some experiments either on a real dataset that has the cold start problem or evaluate different scenarios in existing datasets. Also it is not clear why when all features are removed why nodes which are further away are more important.

3-  As mentioned in the paper the slope of the Relu affects the oversmoothing. That would be interesting how varying such slopes would affect the performance. It is mentioned that a large slope will cause exploding gradients. But I am wondering what the trend looks like as we increase the slope. Also can we say if the slope is smaller than one the oversmoothing happens earlier and all representations converge to a similar subspace?


**Summary Of The Paper:**

The paper consider the oversmoothing problem in deep GNNs and it showed activation function plays a crucial role in this phenomena. The paper proposes a simple modification to the slope of ReLU to reduce oversmoothing.  Their experimental results showed improvement on deep GNNs on classification tasks even though still a shallow GNNs works better than the deep network. They showed that deep GNNs, which do not suffer from oversmoothing, can be beneficial in the presence of the cold start problem.


**Summary Of The Review:**

In general the theoretical analysis the paper provides sounds interesting and it sheds more light on controllable parameters in GNNs to delay oversmoothness but still it is not clear how useful it is in practice since shallow GNNs are still better real scenarios.

---

> ### Author Response · Authors · 2022-11-15
> **Response to Reviewer rfT6**
>
> Thank you for the comments and helpful feedback on our work. **We want to inform the reviewers that the manuscript has been updated including an extended experimental subsection, Appendix F.** The aim of the paper is to show that a simple method, applicable to many architectures can reduce the problem of oversmoothing.
>
> **Usefulness in a real scenario.** We agree with the reviewer that shallow graph NNs perform better in many classification tasks and we share the same concerns regarding the situations where deep graph NNs are needed. That was the main reason why we have included the “cold start” problem, which is a hard problem where deep architectures provide some benefit. The aim of the paper is to demonstrate how a small change in the slope of the activation function can reduce oversmoothing and make deep architectures usable where needed. There are also more complex methods to address the problem of oversmoothing (e.g. using residual connections), but our claim is that one can go a long way by this very simple modification. For completeness, we have included in the new Appendix F experiments with a residual architecture (GCNII).
>
> **Validity of the “cold start” problem.** In order to simulate the “cold start” scenario, we have removed feature vectors, following the method proposed  in (Pairnorm, Zhao & Akoglu, 2020). That particular setup is suitable for deep architectures, but models prone to oversmoothing fail. We agree with the reviewer that a real-world “cold start” dataset would be beneficial, but we have not been able to locate a suitable benchmark.
>
> **Why cold start makes distant nodes more important.** When feature vectors are replaced with all-zero vectors the only nodes containing meaningful representations are the labeled ones. Thus, some nodes (not directly connected with labeled nodes) would have to use multi hop neighborhoods, in order to reach the labeled nodes and create informative representations.
>
> **Slope sensitivity.** We agree with the reviewer regarding the need to show the results for different slope values and we have included them in Appendix F. The results confirm the choice of slope=2 and the method does not seem to be sensitive to the exact value that is chosen. Moreover, to better illustrate the effect of changing the slope to the representations of the nodes, we have included results of t-SNE (van der Maaten and Hinton, 2008) with slope 1 and slope 2 in Appendix F.
>
> **Validity of assumption 2.** This is indeed a very interesting issue. We have tried to make it as general as possible in order to hold for GNNs, which are indeed more restricted than FFNNs. The differences between the two architectures and their implications on the assumption are mentioned on page 4 of the main text, following the introduction of the assumption. GNN’s restrictive nature compared to FFNNs was the main reason why we have not used the probability equal to 1/2  as proposed in (Lu et. al 2019). Instead of that we have shown that our analysis (lemma 2 and theorem 2) holds as long as this probability is bounded from above by any constant p $\in$ [0, 1). This upper bound is ensured by assumption 2.

---

### Official Review · Reviewer_WALS · 2022-10-26

**Confidence:** 3
**Correctness:** 4
**Technical Novelty And Significance:** 3
**Empirical Novelty And Significance:** 3
**Recommendation:** 6

**Clarity, Quality, Novelty And Reproducibility:**

The paper is clearly written. Paper is of good quality. It is novel in its analysis as well as associated emperical study. Experiments are well explained and I consider the work reproducible.

**Details Of Ethics Concerns:**

I see no issues

**Strength And Weaknesses:**

**Strengths**

1. The main idea of the paper is intuitive and simple, and can easily be incorporated in graph NNs.
2. The idea is theoreticall well motivated, and experimentally validated.
3. Although I did not check all proofs, most are based on existing results and the proof outlines sketched in the main text make sense, as such I believe them to be correct.

**Weaknesses**

1. The paper is, understandably, limited to graph NNs of the convolutional kind (based on spectral convolutions). Given the popularity of other graph NN approaches (s.a. message passing, transformers, ...) it would be interesting to learn how these results generalize to other type of graph NNs.
2. Section 4.1, models. The paper uses relatively basic architectures, which is important to ablate the effect of the modified activation function, however, it would still be nice to have models with residual connections just to have a baseline. It would give insight on how important it actually is to go deep with only basic building blocks.
3. The paper addresses the issue of oversmoothing, but only quantifies it indirectly be comparing test accuracies. Wouldn't it be possible to quantify the heterogeneity of features in graphs somehow, as a function of depth? E.g. via metrics present in this work:
*Chen, D., Lin, Y., Li, W., Li, P., Zhou, J., & Sun, X. (2020, April). Measuring and relieving the over-smoothing problem for graph neural networks from the topological view. In Proceedings of the AAAI Conference on Artificial Intelligence (Vol. 34, No. 04, pp. 3438-3445).*
4. Small typo: page 4 "We use the notation of 'converging to a subspace'" -> "... notion of ..."

**Summary Of The Paper:**

The paper analyses the phenomenon of oversmoothing in graph convolutional neural networks based on the graph laplacian. It propose a simple, but theoretically well-motivated, and experimentally well-validated solution to this problem. The solution builds upon previous theoretical analyses on how the variances in feature maps decreases with each graph convolution layer. Bounds are provided on how quickly this happens based on the singular values of the weight matrices. The theoretical analysis shows that one should simply scale the slope in ReLU activation functions with a factor 2. This solution greatly reduces the speed at which feature values converge to the oversmoothed sub-space in which feature eventually land. Thus, the simple trick allows to build deeper graph NNs.

**Summary Of The Review:**

Considering the simplicity of the proposed solution, and the fact that graph convolutions are still very much prevalent in graph NN literature, the paper could have a high impact. Considering that the paper is also well written and sound, I recommend accept.

---

> ### Author Response · Authors · 2022-11-15
> **Response to Reviewer WALS**
>
> Thank you for the comments and helpful feedback on our work. **We want to inform the reviewers that the manuscript has been updated including an extended experimental subsection, Appendix F.** The aim of the paper is to show that a simple method, applicable to many architectures can reduce the problem of oversmoothing.
>
> **Do our results generalize to other architectures?** The aim of this work was to present a simple method that could be utilized in many graph NNs. Nevertheless, it would indeed be interesting for someone to further explore the applicability of the method to other NN approaches.
>
> **Using residual GNNs as baselines.** The basic claim of our work is that methods that suffer from the problem of oversmoothing can be improved significantly in a very simple way. Residual architectures deal with  oversmoothing in a more complex manner. For reasons of completeness, we have extended our experiments to compare our results against residual architectures. The updated manuscript includes results of GCNII (Chen et. al, 2020) in Appendix F, which extend those in Table 1 and Table 2.
>
> **Measuring oversmoothing otherwise than test accuracies.** We agree that assessing oversmoothing solely through test accuracies can be misleading. Hence, we have analyzed the effect of oversmoothing on node representations, using  t-SNE (van der Maaten and Hinton, 2008). This method allows better visualization of node representations illustrating their heterogeneity and how nodes of different classes move in space. In Appendix F, we have included t-SNE results with slope 1 and slope 2 forGCN and GAT and for [2,4,8,16] layer models. The Figures show graphically how slope 2 reduces oversmoothing and maintains heterogeneous node representations as depth increases.
>
> The authors want to further inform about the inclusion of slope sensitivity experiments in Appendix F, showing that the choice of  slope 2 is a robust one.

---

> > ### Comment · Reviewer_WALS · 2022-11-21
> > **Thank you for the extra work**
> >
> > Dear authors,
> >
> > thank you for your response and the additional figures in appendix F, which I think are insightful and improve the paper. I do intent to stick to my original review score of 6. I.m.o. the paper is worth accepting based on its overall quality, however, some its potential impact might be debatable (concerns by other reviewers) considering the limited scope to spectral convolutions and that other methods suffer less from oversmoothing. Nevertheless, I do not think a paper should be rejected based on *potential" lack of impact, hence still borderline accept, and I do believe spectral methods are ubiquitous enough for this paper to be relevant.

---

> > > ### Author Response · Authors · 2022-11-25
> > > **Thank you for the comments**
> > >
> > > Thank you for the comments on the updated manuscript.
> > >
> > > We agree with the comments regarding ubiquity of spectral convolution methods.
> > >
> > > We hope we have addressed all of the other concerns.

---

### Public Comment · ~Sitao_Luan1 · 2022-11-14
**Relevant Work**

Thank the authors for having this interesting paper which discuss the role of activation functions in over-smoothing problem. I would like to highlight one relevant work [1] that theoretically and empirically studies the effect of activation functions for over-smoothing problem from loss of rank perspective and propose to replace ReLU with Tanh in deep GNNs. Good luck to your rebuttal.

[1] Luan S, Zhao M, Chang X W, et al. Break the ceiling: Stronger multi-scale deep graph convolutional networks[J]. Advances in neural information processing systems, 2019, 32.

---

> ### Author Response · Authors · 2022-11-15
> **Response**
>
> The authors would like to thank you for highlighting that relevant work. It is included in the Related Work section in the updated manuscript.

---

### Decision · Program_Chairs · 2023-01-20

**Decision:**

Reject

**Justification For Why Not Higher Score:**

See meta review for details.

**Justification For Why Not Lower Score:**

Reject

**Metareview: Summary, Strengths And Weaknesses:**

The paper considers the role of activation function in oversmoothing problem in GNNs and provide a remedy for it(change the slope of ReLu activations). They provide theoretical justification for this phenomenon and show the benefits of the proposed method using experiments.

Strength:
- The main idea of the paper is intuitive and simple, and can easily be incorporated in graph NNs.
- The paper developed a simple approach that reduces oversmoothing and enables deep architectures
- The paper provides a theoretical analysis that the convergence to a subspace where node representations are the same can be controlled by two factors 1) slope of the Relu function and 2) modifying the learning rate.


Weakness:
- Even though the paper provides an interesting analysis that correlates the slope of the Relu and the learning rate with the largest singular value of the weight matrix which affects the oversmoothing phenomena, it is still not clear how it is useful in the real scenario given the fact that even their experimental results show that still shallow GNNs performs better in classification task.
- The paper claims that deep GNNs are beneficial in the presence of cold-start problems, but their experimental results sound artificial given that the features of the test data are removed.
- There are multiple new networks that do not suffer from over-smoothing.
- the authors do not provide sufficient investigations into the sensitivity and generalization of the proposed solution. Specifically, several experiments are missing. see reviews for details.